# Improved procedures and computer programs for equivalence assessment of correlation coefficients

**Gwowen Shieh**⊙*

Department of Management Science, National Yang Ming Chiao Tung University, Hsinchu, Taiwan

* gwshieh@nycu.edu.tw

## Abstract

The correlation coefficient is the most commonly used measure for summarizing the magnitude and direction of linear relationship between two response variables. Considerable literature has been devoted to the inference procedures for significance tests and confidence intervals of correlations. However, the essential problem of evaluating correlation equivalence has not been adequately examined. For the purpose of expanding the usefulness of correlational techniques, this article focuses on the Pearson product-moment correlation coefficient and the Fisher's $z$ transformation for developing equivalence procedures of correlation coefficients. Equivalence tests are proposed to assess whether a correlation coefficient is within a designated reference range for declaring equivalence decisions. The important aspects of Type I error rate, power calculation, and sample size determination are also considered. Special emphasis is given to clarify the nature and deficiency of the two one-sided tests for detecting a lack of association. The findings demonstrate the inappropriateness of existing methods for equivalence appraisal and validate the suggested techniques as reliable and primary tools in correlation analysis.

**Data Availability Statement:** All relevant data are within the manuscript and its Supporting information files.

**Funding:** Funding for this project was provided by the Ministry of Science and Technology.

## Introduction

Practical guidelines and suggestions for selecting, calculating, and interpreting effect size indices in statistical analyses have been frequently advocated in the literature. Comprehensive reviews and general principles concerning effect size measures are available in the recent works of Fritz, Morris, and Richler [1], Grissom and Kim [2], Kelley and Preacher [3], Kline [4], Pek and Flora [5], and the references therein. According to the summary in Ferguson [6], effect size measures can fall into four general categories: (1) group difference, (2) strength of association, (3) corrected estimates, and (4) risk estimates. Particularly, Pearson product-moment correlation coefficient or sample correlation coefficient $r$ is the most commonly used strength of association measure in applied research across virtually all disciplines of social sciences. The popularity of sample correlation coefficient in the psychological literature has been documented in de Winter, Gosling and Potter [7], Hemphill [8], and Richard, Bond and Stokes-Zoota [9], among others.

**Competing interests:** The author has declared that no competing interests exist.

Under the normality assumption, the probability density function of the sample correlation coefficient $r$ is extremely complicated as shown in Fisher [10]. Theoretical details and related issues can be found in Chapter 32 of Johnson et al. [11] and Chapter 16 of Stuart and Ord [12]. Exact statistical analyses of the correlation coefficient $\rho$ require complex procedures and involved computation, such as Shieh [13, 14]. To facilitate practical analysis, numerous investigations were devoted to give various expressions, approximations, and computing algorithms for the distribution of the sample correlation coefficient. Notably, the asymptotic normal distributions of the sample correlation coefficient and the Fisher's [15] $z$ transformation have proven to provide reasonable alternatives with satisfying performance in many cases. The intrinsic properties of the Fisher's $z$ transformation in terms of conversion accuracy, geometric interpretation, normalization acceleration, and variance stabilization are demonstrated in Bond and Richardson [16], Hotelling [17], Silver and Dunlap [18], and Winterbottom [19].

It is noteworthy that most presentations of correlational techniques deal primarily with the conventional tests of significance. But methodologists have been strongly advocated to consider replacements for or extensions of the null hypothesis of strict equality to deliver more profound implications in statistical analysis. Specifically, the method of equivalence testing is potentially useful in behavioral and psychological sciences as emphasized in Rogers, Howard, and Vessey [20], Seaman and Serlin [21], Stegner, Bostrom, and Greenfield [22], and Steiger [23]. Meyners [24] presented a discussion of the different types of equivalence tests. Moreover, fundamental principles on the design and analysis of equivalence studies are described in Chow and Liu [25], Hauschke, Steinijans, and Pigeot [26], and Wellek [27].

The two one-sided tests (TOST) procedure of mean equivalence, first described by Schuirmann [28] and Westlake [29], is the most common method in equivalence methodology. Because of the approximate nature, the TOST method possesses conceptual simplicity and computational ease. More importantly, the procedure adequately maintains the Type I error rates and the notion gains general acceptance in practical equivalence problems. Berger and Hsu [30], however, cautioned that the TOST principle may not always preserve the nominal Type I error rates in other circumstances. Within the context of correlation analysis, there are few attempts that study the equivalence testing techniques. The particular case of Goertzen and Cribbie [31] suggested a direct extension of mean equivalence TOST to detect a lack of association. Naturally, the TOST method for assessing the lack of association is presumed to share the same desirable properties of the counterpart TOST for establishing mean equivalence.

It is prudent to note that the lack of association examined in Goertzen and Cribbie [31] concerns what sort of strength of association is so small that it should be described as negligible. It is also constructive and more versatile to evaluate whether a target correlation is close enough to any specific magnitude of substantive interest with respect to the designated equivalence boundaries. The simulation results of Goertzen and Cribbie [31] revealed that the TOST method based on the Fisher's transformation has a serious disadvantage in achieving the nominal Type I error rates. However, no analytic examination and technical illustration have been provided in the literature to elucidate the causes of the problematic behavior. A thorough investigation is required to clarify the nature of such deficiency and its implications for equivalence testing. Goertzen and Cribbie [31] suggested that the detection of a lack of association requires substantially large sample sizes. Monte Carlo simulation methods may give a potential solution to sample size calculation. It is of practical importance to derive the power function and then combine a numerical search to determine the optimal sample sizes.

In view of the importance of equivalence testing and limitations of the current TOST method for correlation coefficients, this paper has four major goals. First, a general framework is considered for appraising correlation equivalence with respect to a designated reference

range that may not be equidistant around the zero value or may not even include the zero value. Therefore, the lack of association is a special case of the presented unified structure. Second, analytic examination and numerical assessment are conducted to illustrate the relative performance of the proposed equivalence procedures. In the process, detailed appraisals and graphic displays are presented to explicate the inherent deficiencies of the TOST method in detecting a lack of association. Third, explicit power functions and sample size algorithms are derived and examined to reveal the exact functional relation and individual impact of the influential factors. They provide researchers a better understanding of the inherent difference that exists between the planned sample sizes conditional on the model configurations. Fourth, it is of practical interest to alleviate the computational demands in equivalence studies. The accompany SAS/IML and R software algorithms are available for conducting the equivalence tests, power calculations, and sample size determinations.

## Methods

Suppose that the paired random variables $(Y_i, X_i)$, $i = 1,\ldots, N$, are independent and identically distributed with bivariate normal distribution with means $\mu_X, \mu_Y$, variances $\sigma_X^2$, $\sigma_Y^2$, and correlation $\rho$. Notably, the correlation coefficient $\rho$ represents an essential effect size measure for the strength of linear relationship between the two variables. The widely used Pearson product-moment correlation coefficient $r$ is a natural estimator for the correlation coefficient $\rho$. It is noteworthy that the normality assumption of $(Y_i, X_i)$, $i = 1,\ldots, N$, provides a convenient and useful setting. However, exact statistical inferences of the correlation coefficient $\rho$ with the sample counterpart $r$ demand considerable analytic and computational complexity. Large-sample approximations are often considered to provide feasible solutions in practical applications.

### Fisher's z transformation

A highly regarded approach to the analysis of population correlation coefficient $\rho$ is based on the famous Fisher's [15] $z$ transformation. Fisher's statistic has an approximately normal distribution

$$\hat{\zeta} = \frac{1}{2}\ln(\frac{1 + r}{1 - r}) \mathrel{\dot\sim} N(\zeta, \sigma_\zeta^2), \tag{1}$$

where $\zeta = ln\{(1 + \rho)/(1 - \rho)\}/2$ and $\sigma_\zeta^2 = 1/(N - 3)$. The large-sample approximations of the sample correlation coefficient $r$ and Fisher's $z$ transformation $\hat{\zeta}$ provide convenient alternatives to correlation assessments. The conventional concerns of correlation analysis focus on the detection of correlation difference with respect to the hypotheses

$$H_0 : \rho = \rho_0 \text{ versus } H_1 : \rho \neq \rho_0,$$

where $\rho_0$ is a chosen quantity. Accordingly, the hypothesis testing can be conducted by rejecting the null hypothesis at the significance level $\alpha$ if $|Z^*| > z_{\alpha/2}$ where $Z^* = (\hat{\zeta} - \zeta_0)/\sigma_\zeta$, $\zeta_0 = ln\{(1 + \rho_0)/(1 - \rho_0)\}/2$, and $z_{\alpha/2}$ is the upper $100(\alpha/2)$-th percentile of the standard normal distribution.

On the other hand, the corresponding large-sample approximation for the distribution of $r$ is

$$r \mathrel{\dot\sim} N(\rho, \sigma_r^2), \tag{2}$$

where $\sigma_r^2 = (1 - \rho^2)^2/(N - 3)$. Fisher's $z$ transformation is largely recommended because the

transformation substantially improves the normality approximation, especially for small sample sizes and extreme sample correlations. Nonetheless, the sample correlation coefficient can still have intrinsic values in specific problems and complex situations such as Olkin and Finn [32, 33] and Steiger [34]. Despite the great interest in correlation analysis, there exist few studies that explicitly address the problem of how to appraise correlation equivalence. With the asymptotic normality properties of the sample correlation coefficient and Fisher's $z$ transformation, extended procedures are proposed for equivalence assessment of correlation coefficients.

## The extended sample correlation coefficient procedure

The primary focus of this article is on the equivalence test of correlation coefficient with respect to the null and alternative hypotheses:

$$H_0 : \rho \leq \rho_L \text{ or } \rho_U \leq \rho \text{ versus } H_1 : \rho_L < \rho < \rho_U, \tag{3}$$

where $\rho_L$ and $\rho_U$ are two constants that $(\rho_L, \rho_U)$ represents the designated range for declaring equivalence. Related discussions for selecting a specific margin or threshold for equivalence research are available in Piaggio et al. [35], Walker and Nowacki [36], and Wiens [37]. The general theorem to deriving optimal parametric tests for equivalence hypotheses was presented in Wellek [27], Section 3.3. Also, the determination of rejection region of the optimal procedure follows from the general results in Lehmann and Romano [38], Section 3.4, for tests in families with monotone likelihood ratio. To claim the population correlation $\rho$ is within the interval $(\rho_L, \rho_U)$, a natural rejection region to the null hypothesis is

$$\text{EQUT} - r = \{\hat{r}_{EQUT.L} < r < \hat{r}_{EQUT.U}\}, \tag{4}$$

where the two critical values $\hat{r}_{EQUT.L}$ and $\hat{r}_{EQUT.U}$ are chosen to simultaneously attain the nominal Type I error rate

$$P\{\hat{r}_{EQUT.L} < r < \hat{r}_{EQUT.U} \mid \rho = \rho_L\} = \alpha \text{ and } P\{\hat{r}_{EQUT.L} < r < \hat{r}_{EQUT.U} \mid \rho = \rho_U\} = \alpha. \tag{5}$$

Due to the complexity of the exact distribution function of $r$, the asymptotic normal distribution $r \dot\sim N(\rho, \sigma_r^2)$ given in Eq 2 is a feasible method. Thus, the two probabilities $P\{\hat{r}_{EQUT.L} < r < \hat{r}_{EQUT.U} \mid \rho = \rho_L\}$ and $P\{\hat{r}_{EQUT.L} < r < \hat{r}_{EQUT.U} \mid \rho = \rho_U\}$ can be evaluated by the approximate normal distributions $r \dot\sim N(\rho_L, \sigma_L^2)$ and $r \dot\sim N(\rho_U, \sigma_U^2)$, respectively, where $\sigma_L^2 = (1 - \rho_L^2)^2/(N - 3)$ and $\sigma_U^2 = (1 - \rho_U^2)^2/(N - 3)$. Note that the two quantities $\hat{r}_{EQUT.L}$ and $\hat{r}_{EQUT.U}$ are functions of the configurations $\{\alpha, N, \rho_L, \rho_U\}$. Essentially, they have no explicit analytic expression and require a computer program to calculate the actual values.

## The extended Fisher's z transformation procedure

In view of the widely used Fisher's transformation for correlation analysis, an alternative approach to assessing correlation equivalence is testing the null and alternative hypotheses:

$$H_0 : \zeta \leq \zeta_L \text{ or } \zeta_U \leq \zeta \text{ versus } H_1 : \zeta_L < \zeta < \zeta_U, \tag{6}$$

where $\zeta_L = ln\{(1 + \rho_L)/(1 - \rho_L)\}/2$, $\zeta_U = ln\{(1 + \rho_U)/(1 - \rho_U)\}/2$. Accordingly, the interval $(\zeta_L, \zeta_U)$ indicates the designated bounds for declaring equivalence with respect to the transformed parameter $\zeta$. In this case, the rejection region is of the form

$$\text{EQUT} - \hat{\zeta} = \{\hat{\zeta}_{EQUT.L} < \hat{\zeta} < \hat{\zeta}_{EQUT.U}\}, \tag{7}$$

where the two critical values $\hat{\zeta}_{EQUT.L}$ and $\hat{\zeta}_{EQUT.U}$ simultaneously achieve the nominal Type I error rate

$$P\{\hat{\zeta}_{EQUT.L} < \hat{\zeta} < \hat{\zeta}_{EQUT.U} \mid \zeta = \zeta_L\} = \alpha \text{ and } P\{\hat{\zeta}_{EQUT.L} < \hat{\zeta} < \hat{\zeta}_{EQUT.U} \mid \zeta = \zeta_U\} = \alpha. \quad (8)$$

Following the accurate approximation of $\hat{\zeta} \dot{\sim} N(\zeta, \sigma_\zeta^2)$ given in Eq 1, the two probabilities $P\{\hat{\zeta}_{EQUT.L} < \hat{\zeta} < \hat{\zeta}_{EQUT.U} \mid \zeta = \zeta_L\}$ and $P\{\hat{\zeta}_{EQUT.L} < \hat{\zeta} < \hat{\zeta}_{EQUT.U} \mid \zeta = \zeta_U\}$ can readily be evaluated by the approximate normal distributions $\hat{\zeta} \dot{\sim} N(\zeta_L, \sigma_\zeta^2)$ and $\hat{\zeta} \dot{\sim} N(\zeta_U, \sigma_\zeta^2)$, respectively. For ease of application, the rejection region EQUT-$\hat{\zeta}$ is commonly converted into the scale of $r$ by the conversion formula $r = (e^{2\hat{\zeta}} - 1)/(e^{2\hat{\zeta}} + 1)$. Thus, a useful expression of EQUT-$\hat{\zeta}$ is

$$\text{EQUT} - \hat{\zeta} = \{r(\hat{\zeta}_{EQUT.L}) < r < r(\hat{\zeta}_{EQUT.U})\}, \quad (9)$$

where $r(\hat{\zeta}_{EQUT.L}) = (e^{2\hat{\zeta}_{EQUT.L}} - 1)/(e^{2\hat{\zeta}_{EQUT.L}} + 1)$ and $r(\hat{\zeta}_{EQUT.U}) = (e^{2\hat{\zeta}_{EQUT.U}} - 1)/(e^{2\hat{\zeta}_{EQUT.U}} + 1)$.

Under the asymptotic theory, Fisher's transformation has vital implications in normalization acceleration and variance stabilization relative to the sample correlation coefficient. The discrepancy between the two equivalence approaches with the designated rejection regions EQUT-$r$ and EQUT-$\hat{\zeta}$ will be explicated in the subsequent numerical illustrations.

## Numerical examples

The summary of Hemphill [8] revealed that approximately one third of the correlation coefficients are less than 0.20, one third fall between 0.20 and 0.30, and one third are more than the magnitude 0.30 in the research literature of psychological assessment and treatment. Also, the comprehensive review of Richard et al. [9] showed that the average magnitude of correlation coefficients in psychological literature is 0.21. Accordingly, only the values between 0 and 0.3 are evaluated for the reference bounds $\rho_L$ and $\rho_U$ in the numerical illustration. With the significance level $\alpha = 0.05$, the rejection regions of the two equivalence tests are computed for the reference range $(\rho_L, \rho_U) = (0, 0.20)$, $(0.05, 0.15)$, $(0.10, 0.30)$, and $(0.15, 0.25)$ and sample size $N = 25, 50, 100$, and 500.

Simulation study of 10,000 iterations was also conducted to assess the accuracy of rejection regions through the differences between the simulated Type I error rate and the nominal alpha level 0.05. The associated results of rejection regions and simulation errors are summarized in Table 1. Although both test procedures are constructed under asymptotic theory, they achieve nearly the specified Type I error rate even for small sample sizes $N = 25$ and 50. To visualize the similarities and differences between the two procedures, the rejection regions for

**Table 1. The critical intervals and simulated errors of the suggested correlation equivalence tests for $\alpha = 0.05$.**

| | N | 25 | | 50 | | 100 | | 500 | |
|---|---|---|---|---|---|---|---|---|---|
| $(\rho_L, \rho_U)$ | Procedure | (L, U) | Error | (L, U) | Error | (L, U) | Error | (L, U) | Error |
| (0.00, 0.20) | EQUT-$r$ | (0.0784, 0.1079) | −0.0045 | (0.0864, 0.1093) | −0.0019 | (0.0897, 0.1104) | −0.0042 | (0.0730, 0.1298) | 0.0029 |
| | EQUT-$\hat{\zeta}$ | (0.0862, 0.1158) | −0.0046 | (0.0895, 0.1125) | −0.0007 | (0.0906, 0.1114) | −0.0019 | (0.0728, 0.1290) | 0.0013 |
| (0.05, 0.15) | EQUT-$r$ | (0.0780, 0.1051) | −0.0011 | (0.0867, 0.1059) | 0.0035 | (0.0913, 0.1056) | −0.0030 | (0.0949, 0.1054) | −0.0037 |
| | EQUT-$\hat{\zeta}$ | (0.0866, 0.1138) | −0.0002 | (0.0906, 0.1099) | 0.0020 | (0.0931, 0.1074) | −0.0037 | (0.0950, 0.1055) | −0.0034 |
| (0.10, 0.30) | EQUT-$r$ | (0.1727, 0.2015) | −0.0068 | (0.1849, 0.2075) | −0.0037 | (0.1899, 0.2106) | −0.0033 | (0.1725, 0.2332) | 0.0044 |
| | EQUT-$\hat{\zeta}$ | (0.1876, 0.2165) | −0.0037 | (0.1907, 0.2134) | −0.0018 | (0.1917, 0.2125) | −0.0037 | (0.1719, 0.2320) | 0.0020 |
| (0.15, 0.25) | EQUT-$r$ | (0.1705, 0.1970) | 0.0017 | (0.1836, 0.2023) | 0.0002 | (0.1901, 0.2041) | −0.0027 | (0.1950, 0.2056) | −0.0043 |
| | EQUT-$\hat{\zeta}$ | (0.1873, 0.2137) | 0.0014 | (0.1911, 0.2099) | 0.0035 | (0.1935, 0.2075) | 0.0016 | (0.1952, 0.2058) | −0.0041 |

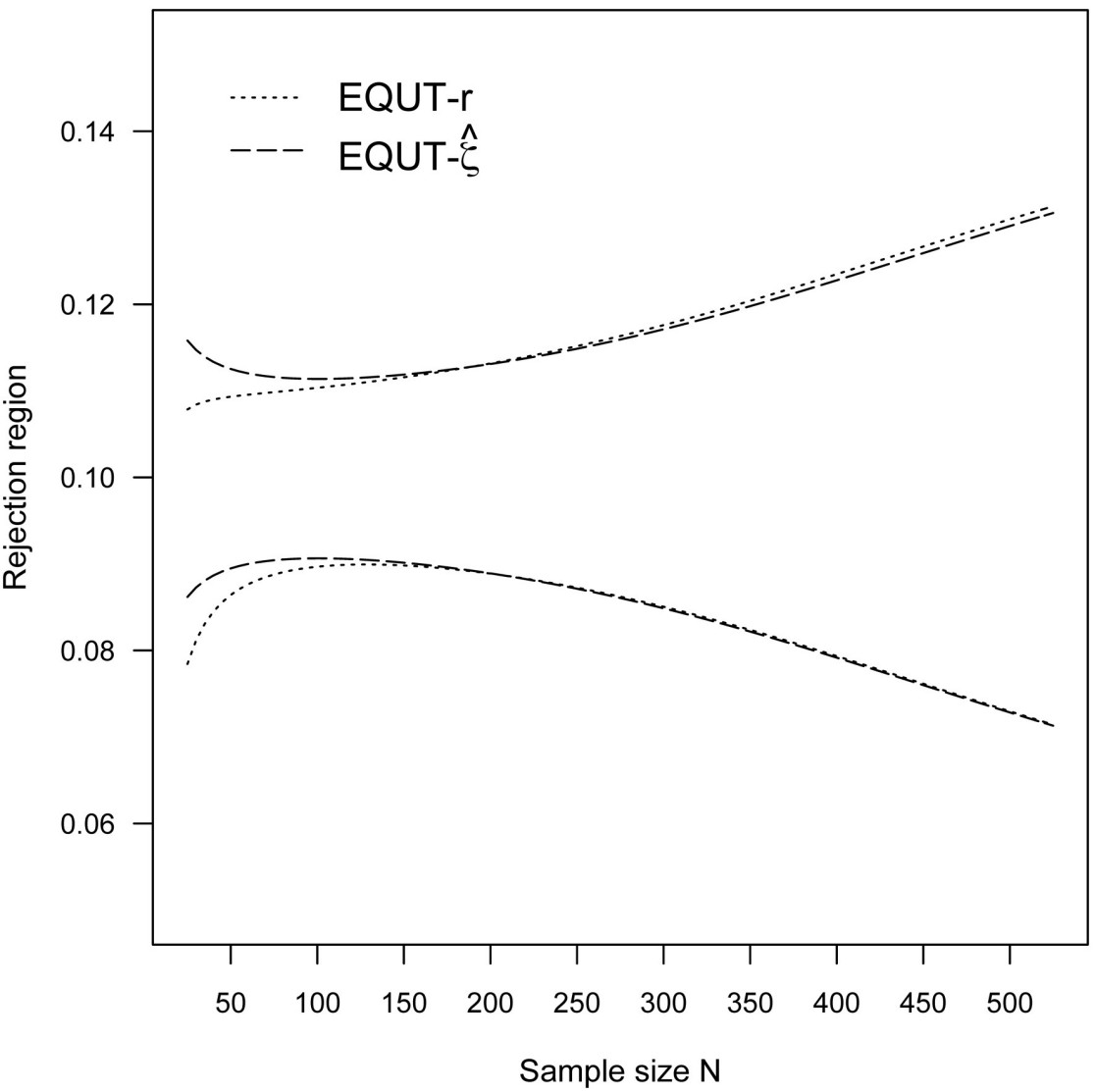

**Fig 1. The rejection regions for $(\rho_L, \rho_U) = (0, 0.2)$ and $\alpha = 0.05$.**

$(\rho_L, \rho_U) = (0, 0.20)$ and $(0.10, 0.30)$ are also plotted in Figs 1 and 2, respectively. The rejection regions of the two procedures have distinct outcomes for small sample sizes $N < 150$ and are nearly identical for larger sample sizes $N \geq 150$.

## Results

An important scenario in equivalence assessment is the detection of a lack of association or the population correlation $\rho$ is practically zero. Accordingly, the asymptotic normal distributions of the simple correlation $r$ and the associated transformation $\hat{\zeta}$ have zero mean when the population correlation $\rho = 0$. Due to the symmetric feature of normal distributions for the two principal statistics, it is sensible to adopt an equidistant reference range about zero in assessing the lack of association. Thus, the problem of probing a lack of association can be viewed as a special setting of the proposed general framework for correlation equivalence detection.

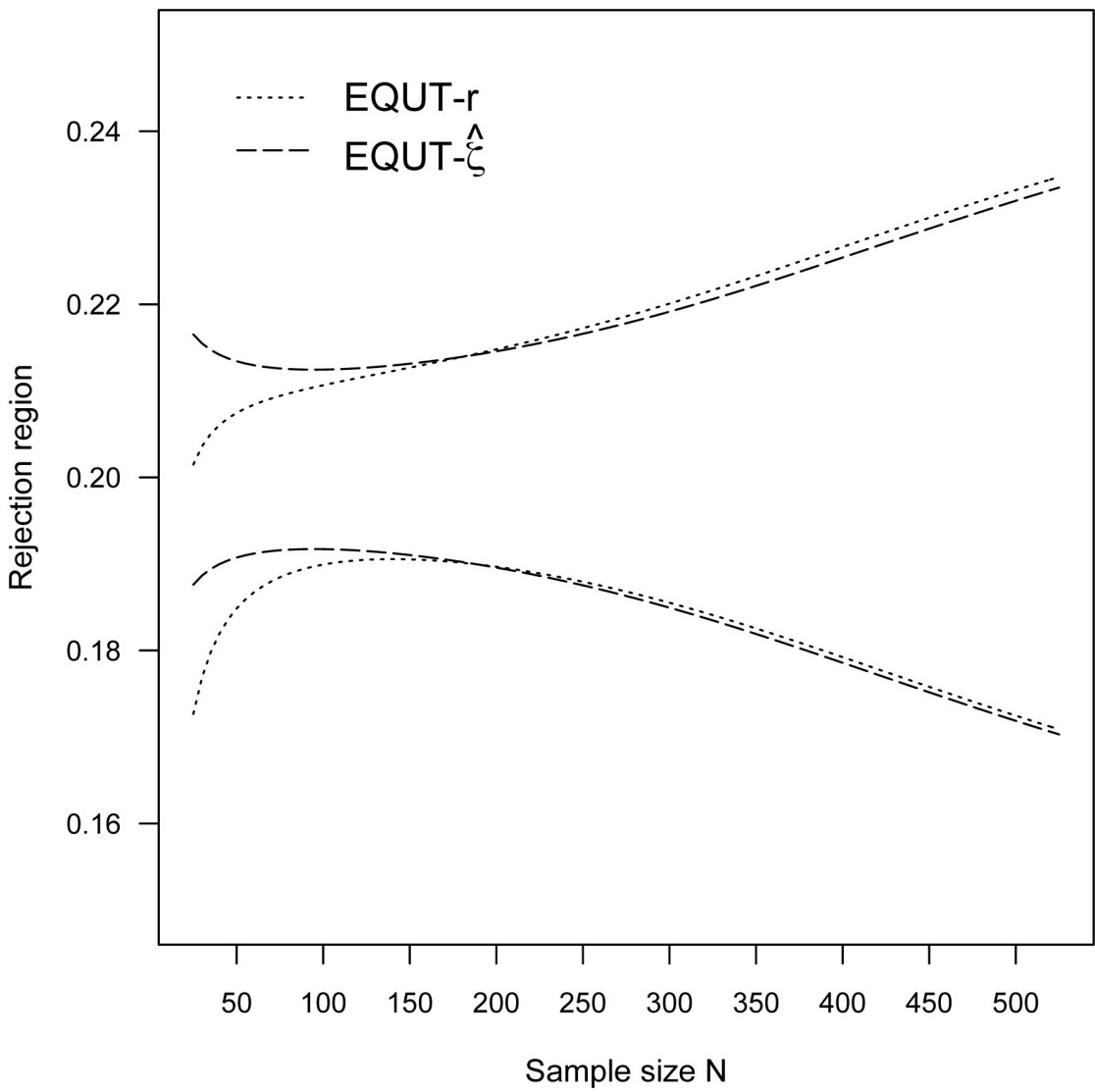

**Fig 2. The rejection regions for ($\rho_L$, $\rho_U$) = (0.1, 0.3) and $\alpha$ = 0.05.**

### The proposed lack of association tests

To examine the lack of association, the prescribed hypotheses for equivalence testing are readily modified with $\rho_L = -\rho_B$ and $\rho_U = \rho_B$ with $\rho_B > 0$:

$$\text{H}_0 : \rho \leq -\rho_B \text{ or } \rho_B \leq \rho \text{ versus H}_1 : -\rho_B < \rho < \rho_B, \tag{10}$$

where the designated bound $\rho_B$ indicates the maximal tolerance magnitude to claim a lack of association. The equivalence procedures based on the two statistics $r$ and $\hat{\zeta}$ can immediately be applied to the current problem for testing a lack of association.

With the symmetric equivalence range ($-\rho_B$, $\rho_B$) around zero, the subsequent explication shows that two critical values $\hat{r}_L$ and $\hat{r}_U$ of the prescribed equivalence procedure have a simple relation $\hat{r}_U = -\hat{r}_L$. Note that the sample correlation coefficient $r$ has the approximate distribution $N(-\rho_B, \sigma_B^2)$ and $N(\rho_B, \sigma_B^2)$ for $\rho = -\rho_B$ and $\rho_B$, respectively, where

$\sigma_B^2 = (1 - \rho_B^2)^2/(N - 3)$. Hence, the approximate distribution of $-r \dot\sim N(\rho_B, \sigma_B^2)$ under $\rho = -\rho_B$ coincides that of $r \dot\sim N(\rho_B, \sigma_B^2)$ under $\rho = \rho_B$. As described earlier, the actual values $\hat{r}_L$ and $\hat{r}_U$ are uniquely determined by the two probabilities $P\{\hat{r}_L < r < \hat{r}_U \mid \rho = -\rho_B\} = \alpha$ and $P\{\hat{r}_L < r < \hat{r}_U \mid \rho = \rho_B\} = \alpha$. The normal approximation of $r$ implies the former equality is closely related to the latter:

$$P\{\hat{r}_L < r < \hat{r}_U \mid \rho = -\rho_B\} = P\{-\hat{r}_U < -r < -\hat{r}_L \mid \rho = -\rho_B\} = P\{-\hat{r}_U < r < -\hat{r}_L \mid \rho = \rho_B\} = \alpha.$$

Accordingly, this examination establishes that $\hat{r}_U = -\hat{r}_L$ and the rejection region can be simplified as

$$\text{EQUT} - r = \{-\hat{r}_{EQUT} < r < \hat{r}_{EQUT}\}, \tag{11}$$

where $\hat{r}_{EQUT}$ is chosen so that

$$P\{-(\hat{r}_{EQUT} + \rho_B)/\sigma_B < Z < (\hat{r}_{EQUT} - \rho_B)/\sigma_B\} = \alpha \tag{12}$$

and $Z = (r - \rho_B)/\sigma_B \dot\sim N(0, 1)$.

Under the notion of Fisher transformation, the lack of association test can alternatively be conducted in terms of the hypotheses:

$$H_0 : \zeta \leq -\zeta_B \text{ or } \zeta_B \leq \zeta \text{ versus } H_1 : -\zeta_B < \zeta < \zeta_B, \tag{13}$$

where $\zeta_B = \ln\{(1 + \rho_B)/(1 - \rho_B)\}/2$. Following the arguments similar to the previous case for $r$, the rejection region for the transformed test statistic $\hat{\zeta}$ is of the form

$$\text{EQUT} - \hat{\zeta} = \{-\hat{\zeta}_{EQUT} < \hat{\zeta} < \hat{\zeta}_{EQUT}\}, \tag{14}$$

where the quantity $\hat{\zeta}_{EQUT}$ satisfies

$$P\{-(\hat{\zeta}_{EQUT} + \zeta_B)/\sigma_\zeta < Z < (\hat{\zeta}_{EQUT} - \zeta_B)/\sigma_\zeta\} = \alpha \tag{15}$$

and $Z = (\hat{\zeta} - \zeta_B)/\sigma_\zeta \dot\sim N(0, 1)$. The rejection region EQUT-$\hat{\zeta}$ can also be transformed into an interval on $r$ as

$$\text{EQUT} - \hat{\zeta} = \{-r(\hat{\zeta}_{EQUT}) < r < r(\hat{\zeta}_{EQUT})\}, \tag{16}$$

where $r(\hat{\zeta}_{EQUT}) = (e^{2\hat{\zeta}_{EQUT}} - 1)/(e^{2\hat{\zeta}_{EQUT}} + 1)$.

## Two one-sided tests procedures

With the popular mean equivalence TOST procedure of Schuirmann [28] and Westlake [29], it is temping to generalize the appealing principle for correlation evaluation with the sample correlation $r$ and the transformation $\hat{\zeta}$. Using the asymptotic normal distribution of $r$, a TOST procedure for detecting a lack of association can easily be constructed with the approximate normal distribution $r \dot\sim N(\rho_B, \sigma_B^2)$. Specifically, the null hypothesis H$_0$: $\rho \leq -\rho_B$ or $\rho_B \leq \rho$ is rejected at the significance level $\alpha$ if

$$R_L = \frac{r + \rho_B}{\sigma_B} > z_\alpha \text{ and } R_U = \frac{r - \rho_B}{\sigma_B} < -Z_\alpha, \cdot \tag{17}$$

where $\sigma_B^2 = (1 - \rho_B^2)^2/(N - 3)$ and $z_\alpha$ is the upper $100\,\alpha$-th percentile of the standard normal distribution. For ease of explication, the procedure is termed as the TOST-$r$ test and the

associated rejection region is expressed as

$$\text{TOST} - r = \{-\hat{r}_{Tost} < r < \hat{r}_{TOST}\} \tag{18}$$

where $\hat{r}_{TOST} = \rho_B - Z_\alpha \sigma_B$. Regarding the Type I errors, the TOST-$r$ procedure should attain the nominal alpha level when $\rho = \rho_B$ or $\rho = -\rho_B$. Accordingly, the true Type I error rate of TOST-$r$ is

$$
\begin{aligned}
P\{-\hat{r}_{TOST} < r < \hat{r}_{TOST} \mid \rho = -\rho_B\} &= P\{-\hat{r}_{TOST} < r < \hat{r}_{TOST} \mid \rho = \rho_B\} \\
&= P\{z_\alpha - 2\rho_B/\sigma_B < Z < -Z_\alpha\},
\end{aligned}
\tag{19}
$$

where $Z = (r - \rho_B)/\sigma_B \dot{\sim} N(0, 1)$.

Similarly, a TOST procedure can be obtained with the Fisher's transformation for detecting a lack of association as previously suggested by Goertzen and Cribbie [31]. This procedure is denoted by TOST-$\hat{\zeta}$ and it rejects the null hypothesis $H_0$: $\zeta \leq -\zeta_B$ or $\zeta_B \leq \zeta$ at the significance level $\alpha$ if

$$Z_L = \frac{\hat{\zeta} + \zeta_B}{\sigma_\zeta} > z_\alpha \text{ and } Z_U = \frac{\hat{\zeta} - \zeta_B}{\sigma_\zeta} < -Z_\alpha. \tag{20}$$

The resulting rejection region can also be written as:

$$\text{TOST} - \hat{\zeta} = \{-\hat{\zeta}_{TOST} < \hat{\zeta} < \hat{\zeta}_{TOST}\} \tag{21}$$

where $\hat{\zeta}_{TOST} = \zeta_B - Z_\alpha \sigma_\zeta$. Moreover, the asymptotic distribution of $\hat{\zeta}$ reveals that the true Type I error rate is

$$
\begin{aligned}
P\{-\hat{\zeta}_{TOST} < \hat{\zeta} < \hat{\zeta}_{TOST} \mid \zeta = -\zeta_B\} &= P\{-\hat{\zeta}_{TOST} < \hat{\zeta} < \hat{\zeta}_{TOST} \mid \zeta = \zeta_B\} \\
&= P\{z_\alpha - 2\zeta_B/\sigma_\zeta < Z < -Z_\alpha\},
\end{aligned}
\tag{22}
$$

where $Z = (\hat{\zeta} - \zeta_B)/\sigma_\zeta \dot{\sim} N(0, 1)$.

## Type I errors

The most important property of a test procedure is to provide acceptable level of Type I errors. Without the adequate or excellence adherence to the nominal $\alpha$ levels, the accompanying power evaluations and statistical assessments are meaningless on the basis of distorted Type I error behavior. It follows from the analytic justifications in Eqs 12 and 15 that the two suggested equivalence procedures EQUT-$r$ and EQUT-$\hat{\zeta}$ have excellent performance in maintaining the nominal Type I error rates. In contrast, the other two TOST counterparts are problematic as explained next.

Note that the (supremum) Type I error rate of the mean equivalence TOST method is exactly equal to the nominal alpha level as the sample size goes to infinity, even though the true rejection probability is less than the designated alpha level for all possible configurations under the null hypothesis. For the direct generalization of TOST-$r$ procedure for correlation equivalence, however, the rejection region TOST-$r$ given in Eq 18 is a proper interval only when $\hat{r}_{TOST} > 0$. It is clear that $\hat{r}_{TOST} > 0$ suggests that $\rho_B > z_\alpha \sigma_B$ or $N > z_\alpha^2 (1 + \rho_B^2)^2 / \rho_B^2 + 3$. Detailed numerical inspections at $\alpha = 0.05$ reveal that TOST-$r$ degenerates as an empty set if $N < 1090.6352$ when $\rho_B = 0.05$, and if $N < 278.9924$ when $\rho_B = 0.10$. To notify this crucial deficiency, the related minimum sample sizes for a nonempty TOST-$r$ are summarized in Table 2 for $\alpha = 0.01$ and 0.05, and $\rho_B = 0.05, 0.10, 0.15, 0.20, 0.25$, and 0.30.

**Table 2. The minimum sample sizes of TOST procedures to have a nonempty critical interval for detecting a lack of association.**

| | α | | | |
|---|---|---|---|---|
| | **0.01** | | **0.05** | |
| $\rho_B$ | **TOST-$r$** | **TOST-$\hat{\zeta}$** | **TOST-$r$** | **TOST-$\hat{\zeta}$** |
| 0.05 | 2179 | 2165 | 1091 | 1084 |
| 0.10 | 556 | 541 | 279 | 272 |
| 0.15 | 255 | 240 | 129 | 122 |
| 0.20 | 150 | 135 | 77 | 69 |
| 0.25 | 101 | 86 | 52 | 45 |
| 0.30 | 75 | 60 | 39 | 32 |

On the other hands, the one-to-one relation between $r$ and $\hat{\zeta}$ implies that the rejection region TOST-$\hat{\zeta}$ shares the same disadvantage as TOST-$r$. The last quantity in Eq 22 indicates that the Type I error rate of the TOST-$\hat{\zeta}$ procedure usually does not attain the nominal level α. However, the Type I error rate of the TOST-$\hat{\zeta}$ method also has the supremum α as the other TOST-$r$ method when the sample size goes to infinity. For finite sample sizes, the rejection region becomes invalid when $\hat{\zeta}_{TOST} = \zeta_B - z_\alpha \sigma_\zeta \leq 0$ or $N \leq z_\alpha^2/\zeta_B^2 + 3$. Specifically, the rejection region TOST-$\hat{\zeta}$ is empty if $N < 1083.4132$ when $\rho_B = 0.05$, and if $N < 271.7488$ when $\rho_B = 0.10$. The minimum sample sizes for a nonempty rejection region TOST-$\hat{\zeta}$ are also listed in Table 2 for α = 0.01 and 0.05, and $\rho_B$ = 0.05, 0.10, 0.15, 0.20, 0.25, and 0.30.

To further demonstrate the fundamental characteristics of the contending equivalence methods, the vital properties of actual Type I error rates are investigated. Specifically, with the significance level α = 0.05, the rejection regions of the four equivalence tests are calculated for the lack of association with the range $(-\rho_B, \rho_B)$ = (–0.1, 0.1) and (–0.2, 0.2), and sample size $N$ = 25, 50, 100, and 500. The rejection regions of the four test procedures for $(-\rho_B, \rho_B)$ = (–0.1, 0.1) and (–0.2, 0.2) are also plotted in Figs 3 and 4, respectively. Moreover, the adequacy of Type I error rate was examined through simulation study of 10,000 iterations and was determined by the deviation between the simulated Type I error rate and the nominal alpha level. The resulting rejection regions and simulation results are listed in Table 3. These numerical evidences suggest that the proposed equivalence procedures EQUT-$r$ and EQUT-$\hat{\zeta}$ have outstandingly performance in achieving the nominal significance level. The two TOST procedures generally do not provide proper rejection regions and adequate levels of Type I errors for small sample sizes, and the situation is more severe when a smaller threshold is considered.

The problematic behavior of TOST-$\hat{\zeta}$ was also demonstrated in the numerical examination (Table 2) of Goertzen and Cribbie [31]. Specifically, their simulation results showed that the resulting Type I error rates of TOST-$\hat{\zeta}$ and two related procedures can be zero for small sample sizes and small correlation bounds. The analytic and empirical findings presented here illustrate the undesirable behavior of the TOST-$r$ and TOST-$\hat{\zeta}$ procedures.

## Power comparisons

The examination of different equivalence procedures further explicates their power behavior for detecting the lack of association through simulation study. With the significance level α = 0.05, the simulated powers of the EQUT-$r$, EQUT-$\hat{\zeta}$, TOST-$r$, and TOST-$\hat{\zeta}$ tests are computed for 10,000 independent samples. The model configurations of correlation coefficient, reference

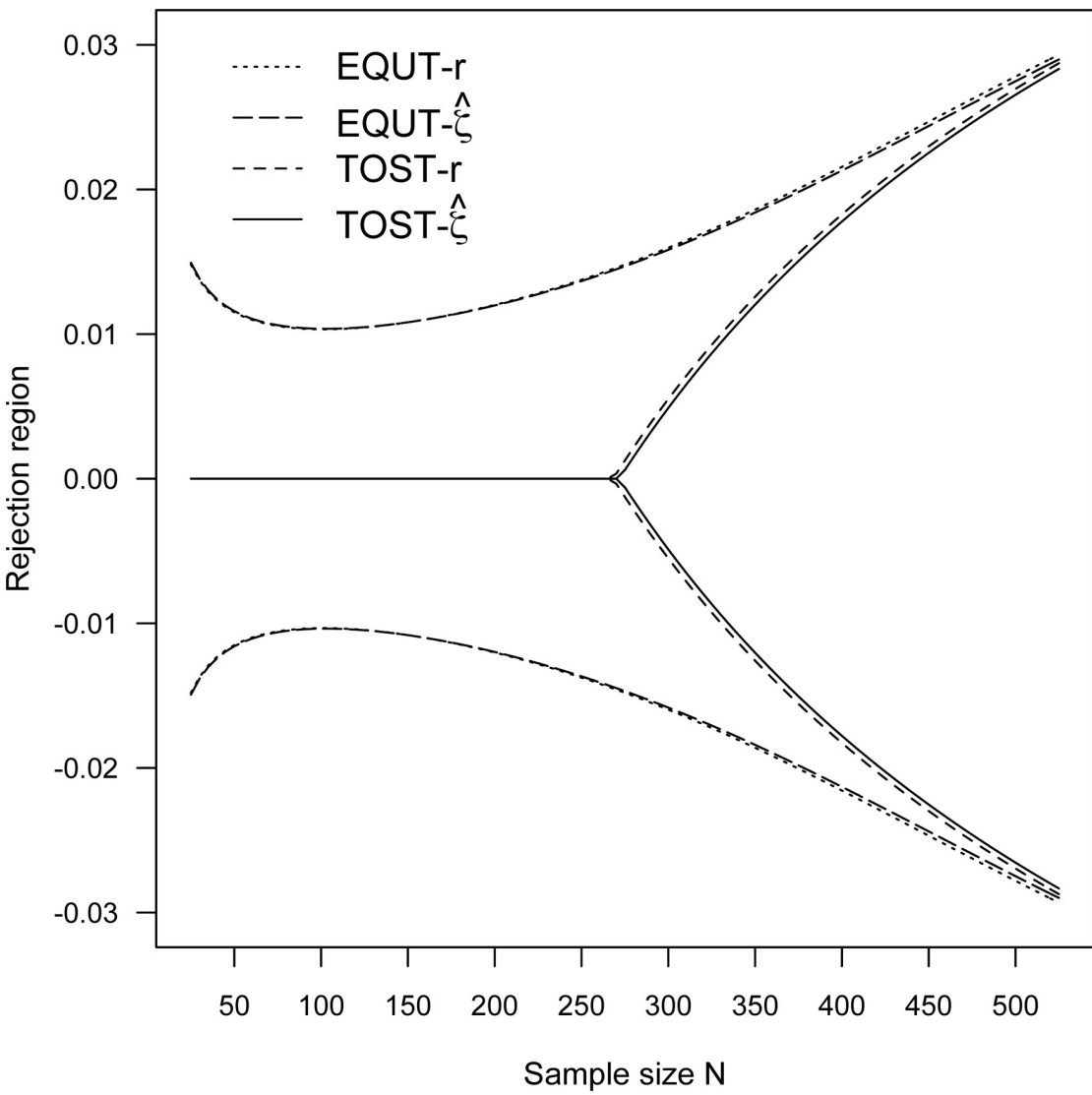

**Fig 3. The rejection regions for $(\rho_L, \rho_U)$ = (-0.1, 0.1) and $\alpha$ = 0.05.**

range, and sample size are chosen as $\rho$ = 0 and 0.05, $(-\rho_B, \rho_B)$ = (−0.1, 0.1) and (−0.2, 0.2), and $N$ = 25, 50, 100, 200, 300, 400, and 500, respectively. The simulated powers of the combined twenty-eight settings are summarized in Table 4 for the four equivalence procedures. The results show that the two suggested procedures have more power than the other two TOST counterparts. Although the differences between these methods diminish for large sample sizes, their discrepancy can be substantial for small and moderate sample sizes. In particular, due to the extremely conservative behavior or the degeneration of rejection region of the two TOST methods, the resulting power values are zero for ten cases in Table 4. For example, both TOST-$r$ and TOST-$\hat{\zeta}$ methods give no power when $(-\rho_B, \rho_B)$ = (−0.1, 0.1) for $N \leq 200$, or when $(-\rho_B, \rho_B)$ = (−0.2, 0.2) for $N \leq 50$. In view of these results, the two TOST procedures are not recommended for detecting a lack of association. The rejection regions EQUT-$r$ and EQUT-$\hat{\zeta}$ assure that the proposed equivalence procedures have superior Type I error rate and power performance.

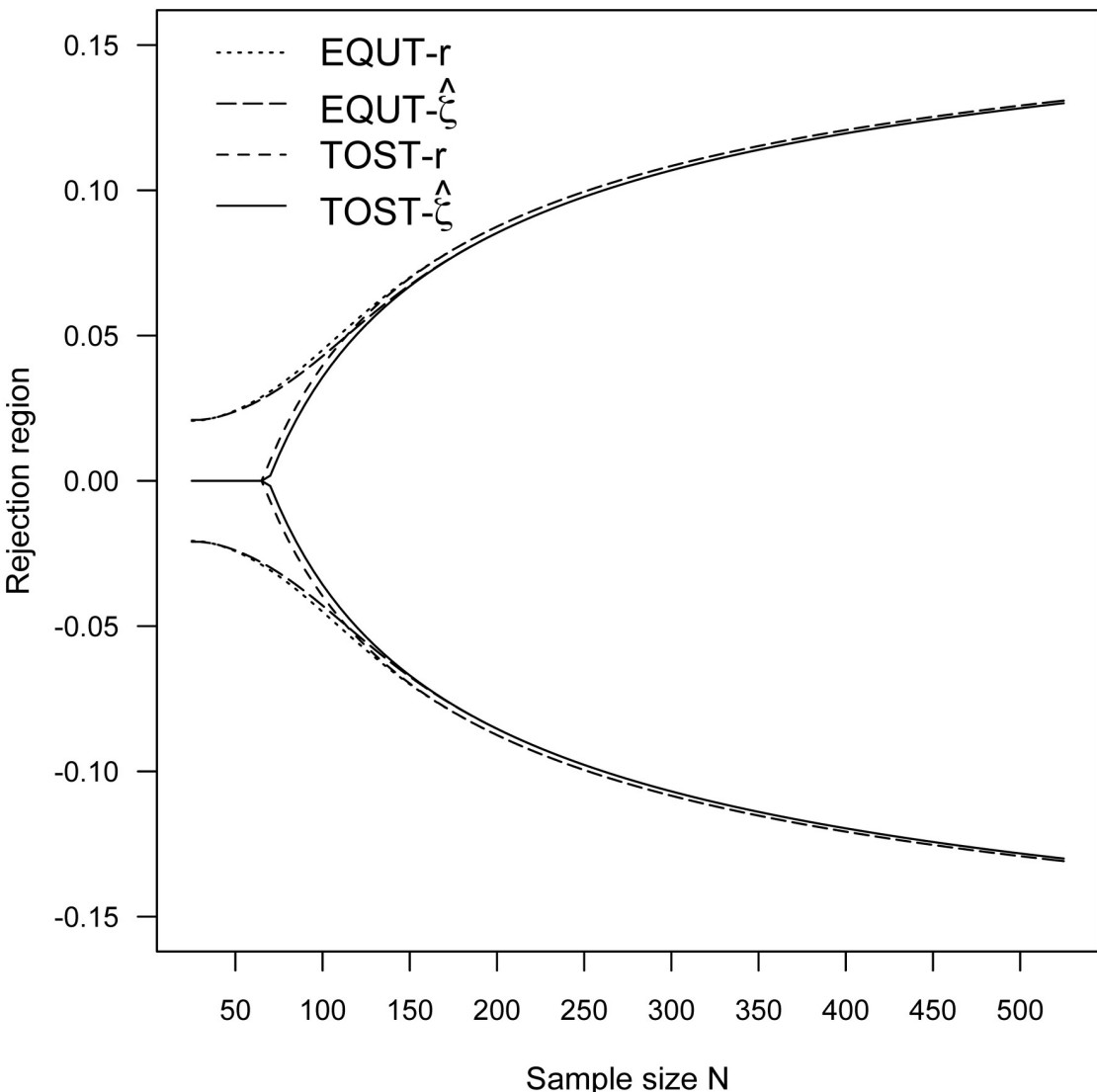

**Fig 4. The rejection regions for $(\rho_L, \rho_U)$ = (-0.2, 0.2) and $\alpha$ = 0.05.**

**Table 3. The critical intervals and simulated errors of the lack of association tests for $\alpha$ = 0.05.**

| | N | 25 | | 50 | | 100 | | 500 | |
|---|---|---|---|---|---|---|---|---|---|
| $(-\rho_B, \rho_B)$ | Procedure | (L, U) | Error | (L, U) | Error | (L, U) | Error | (L, U) | Error |
| (−0.10, 0.10) | EQUT-$r$ | (−0.0148, 0.0148) | −0.0030 | (−0.0115, 0.0115) | 0.0043 | (−0.0103, 0.0103) | 0.0011 | (−0.0278, 0.0278) | 0.0021 |
| | EQUT-$\hat{\zeta}$ | (−0.0149, 0.0149) | −0.0026 | (−0.0116, 0.0116) | 0.0046 | (−0.0104, 0.0104) | 0.0012 | (−0.0275, 0.0275) | 0.0014 |
| | TOST-$r$ | (0, 0) | −0.0500 | (0, 0) | −0.0500 | (0, 0) | −0.0500 | (−0.0270, 0.0270) | 0.0004 |
| | TOST-$\hat{\zeta}$ | (0, 0) | −0.0500 | (0, 0) | −0.0500 | (0, 0) | −0.0500 | (−0.0265, 0.0265) | −0.0010 |
| (−0.20, 0.20) | EQUT-$r$ | (−0.0207, 0.0207) | −0.0020 | (−0.0242, 0.0242) | −0.0020 | (−0.0451, 0.0451) | 0.0010 | (−0.1292, 0.1292) | 0.0034 |
| | EQUT-$\hat{\zeta}$ | (−0.0210, 0.0210) | −0.0012 | (−0.0239, 0.0239) | −0.0026 | (−0.0429, 0.0429) | −0.0021 | (−0.1282, 0.1282) | 0.0022 |
| | TOST-$r$ | (0, 0) | −0.0500 | (0, 0) | −0.0500 | (−0.0397, 0.0397) | −0.0058 | (−0.1292, 0.1292) | 0.0034 |
| | TOST-$\hat{\zeta}$ | (0, 0) | −0.0500 | (0, 0) | −0.0500 | (−0.0357, 0.0357) | −0.0106 | (−0.1282, 0.1282) | 0.0022 |

**Table 4. The simulated powers of the lack of association tests for α = 0.05.**

| (−ρ_B, ρ_B) | ρ | Procedure | N | | | | | | |
|---|---|---|---|---|---|---|---|---|---|
| | | | 25 | 50 | 100 | 200 | 300 | 400 | 500 |
| (−0.10, 0.10) | 0 | EQUT-$r$ | 0.0523 | 0.0613 | 0.0815 | 0.1375 | 0.2202 | 0.3323 | 0.4659 |
| | | EQUT-$\hat{\zeta}$ | 0.0526 | 0.0615 | 0.0817 | 0.1371 | 0.2188 | 0.3283 | 0.4596 |
| | | TOST-$r$ | 0 | 0 | 0 | 0 | 0.0792 | 0.2886 | 0.4501 |
| | | TOST-$\hat{\zeta}$ | 0 | 0 | 0 | 0 | 0.0689 | 0.2818 | 0.4441 |
| (−0.10, 0.10) | 0.05 | EQUT-$r$ | 0.0557 | 0.0608 | 0.0687 | 0.1097 | 0.1502 | 0.2072 | 0.2691 |
| | | EQUT-$\hat{\zeta}$ | 0.0558 | 0.0616 | 0.0690 | 0.1094 | 0.1491 | 0.2037 | 0.2658 |
| | | TOST-$r$ | 0 | 0 | 0 | 0 | 0.0498 | 0.1737 | 0.2596 |
| | | TOST-$\hat{\zeta}$ | 0 | 0 | 0 | 0 | 0.0449 | 0.1677 | 0.2559 |
| (−0.20, 0.20) | 0 | EQUT-$r$ | 0.0762 | 0.1319 | 0.3392 | 0.7740 | 0.9390 | 0.9844 | 0.9964 |
| | | EQUT-$\hat{\zeta}$ | 0.0782 | 0.1307 | 0.3230 | 0.7634 | 0.9349 | 0.9839 | 0.9962 |
| | | TOST-$r$ | 0 | 0 | 0.3022 | 0.7740 | 0.9390 | 0.9844 | 0.9964 |
| | | TOST-$\hat{\zeta}$ | 0 | 0 | 0.2760 | 0.7633 | 0.9349 | 0.9839 | 0.9962 |
| (−0.20, 0.20) | 0.05 | EQUT-$r$ | 0.0754 | 0.1245 | 0.3067 | 0.6707 | 0.8428 | 0.9213 | 0.9634 |
| | | EQUT-$\hat{\zeta}$ | 0.0762 | 0.1229 | 0.2925 | 0.6578 | 0.8357 | 0.9170 | 0.9622 |
| | | TOST-$r$ | 0 | 0 | 0.2729 | 0.6706 | 0.8428 | 0.9213 | 0.9634 |
| | | TOST-$\hat{\zeta}$ | 0 | 0 | 0.2467 | 0.6578 | 0.8357 | 0.9170 | 0.9622 |

## Discussion

A research study requires adequate statistical power and sufficient sample size to examine vital questions and target effects. The importance and implications of statistical power analysis in equivalence testing are also demonstrated in Wellek [27], Murphy, Myros, and Wolach [39], Shieh [40], and Chow et al. [41], among others. To enhance the usefulness of the suggested equivalence procedures, the related issues of power analysis and sample size determination are considered.

### Power and sample size calculations

According to the rejection region EQUT-$r$ defined in Eq 4 of the extended sample correlation procedure, the power function is given by

$$\Psi_r = P\{\hat{r}_{EQUT.L} < r < \hat{r}_{EQUT.U}\} = P\{(\hat{r}_{EQUT.L} - \rho)/\sigma_r < Z < (\hat{r}_{EQUT.U} - \rho)/\sigma_r\}, \quad (23)$$

where $Z = (r - \rho)/\sigma_r \dot{\sim} N(0, 1)$ and $\rho_L < \rho < \rho_U$. Moreover, the rejection region EQUT-$\hat{\zeta}$ defined in Eq 7 of the extended Fisher transformation procedure suggests that the associated power function is of the form

$$\Psi\hat{\zeta} = P\{\hat{r}_{EQUT.L} < \hat{\zeta} < \hat{\zeta}_{EQUT.U}\} = P\{(\hat{r}_{EQUT.L} - \zeta)/\sigma_\zeta < Z < (\hat{\zeta}_{EQUT.U} - \zeta)/\sigma_\zeta\}, \quad (24)$$

where $Z = (\hat{\zeta} - \zeta)/\sigma_\zeta \dot{\sim} N(0, 1)$ and $\zeta_L < \zeta < \zeta_U$. Under the asymptotic normality assumptions, the attained power levels of the two equivalence tests can readily be computed with $\Psi_r$ and $\Psi_{\hat{\zeta}}$ for the specified configurations of equivalence limits $(\rho_L, \rho_U)$, population correlation $\rho$, and significance level α. For advance planning of a research design, the two power formulas can be employed to calculate the sample size $N$ needed to attain the specified power $1 - \beta$ for the chosen significance level α, chosen correlation ρ, and equivalence threshold $(\rho_L, \rho_U)$.

**Table 5. Sample sizes, computed power, and simulated errors of the suggested equivalence tests for nominal power 0.80 and α = 0.05.**

| | Procedure | EQUT-$r$ | | | | EQUT-$\hat{\zeta}$ | | | |
|---|---|---|---|---|---|---|---|---|---|
| $(\rho_L, \rho_U)$ | $\rho$ | $N$ | Simulated power | Estimated power | Error | $N$ | Simulated power | Estimated power | Error |
| (0.0, 0.2) | 0.1 | 834 | 0.8094 | 0.8005 | 0.0089 | 837 | 0.8000 | 0.8002 | −0.0002 |
| (0.1, 0.3) | 0.2 | 785 | 0.8071 | 0.8003 | 0.0068 | 788 | 0.7994 | 0.8006 | −0.0012 |
| (0.2, 0.4) | 0.3 | 708 | 0.8059 | 0.8007 | 0.0052 | 708 | 0.8012 | 0.8002 | 0.0010 |
| (0.3, 0.5) | 0.4 | 606 | 0.8053 | 0.8003 | 0.0059 | 604 | 0.8007 | 0.8000 | 0.0007 |
| (0.4, 0.6) | 0.5 | 488 | 0.8038 | 0.8005 | 0.0033 | 483 | 0.8023 | 0.8002 | 0.0021 |
| (−0.1, 0.1) | 0.0 | 850 | 0.8077 | 0.8002 | 0.0075 | 854 | 0.7971 | 0.8002 | −0.0031 |
| (−0.1, 0.1) | 0.05 | 2440 | 0.7988 | 0.8001 | −0.0013 | 2448 | 0.8049 | 0.8001 | 0.0048 |
| (−0.2, 0.2) | 0.0 | 208 | 0.8098 | 0.8011 | 0.0087 | 212 | 0.7941 | 0.8016 | −0.0075 |
| (−0.2, 0.2) | 0.1 | 585 | 0.8025 | 0.8001 | 0.0024 | 593 | 0.8044 | 0.8002 | 0.0042 |
| (−0.2, 0.2) | 0.05 | 2311 | 0.7995 | 0.8001 | −0.0006 | 2326 | 0.8034 | 0.8000 | 0.0034 |
| (−0.3, 0.3) | 0.0 | 89 | 0.8080 | 0.8013 | 0.0067 | 93 | 0.7974 | 0.8035 | −0.0061 |
| (−0.3, 0.3) | 0.1 | 140 | 0.8006 | 0.8024 | −0.0018 | 145 | 0.8005 | 0.8011 | −0.0006 |
| (−0.3, 0.3) | 0.2 | 535 | 0.8068 | 0.8005 | 0.0063 | 546 | 0.8017 | 0.8005 | 0.0012 |
| (−0.3, 0.3) | 0.25 | 2094 | 0.8011 | 0.8002 | 0.0009 | 2115 | 0.8010 | 0.8000 | 0.0010 |

## Simulation study

Because of the approximate nature of the proposed equivalence procedures, a Monte Carlo simulation study was utilized to appraise the similarities and differences between the suggested power and sample size calculations under a wide variety of correlation configurations. The numerical study was conducted in two steps. First, under the specified settings, the minimum sample sizes required to meet the nominal power 0.80 and α = 0.05 were determined by the power formulas $\Psi_r$ and $\Psi_{\hat{\zeta}}$. The estimated powers or achieved powers are recorded for the optimal sample sizes. Second, with the designated sample sizes, simulated powers were computed with a Monte Carlo simulation study of 10,000 independent data sets to evaluate the accuracy of the two approaches. The accuracy of the two power and sample size procedures is determined by the error between the simulated power and estimated power.

The results of the two procedures EQUT-$r$ and EQUT-$\hat{\zeta}$ are presented in Table 5 for various settings of population correlation ρ, and equivalence range $(\rho_L, \rho_U)$. It can be seen that the optimal sample sizes noticeably vary with the combined characteristics of ρ and $(\rho_L, \rho_U)$. Specifically, when ρ is a varying factor, the sample size increases with decreasing distance = *min* $(\rho_U−\rho, \rho−\rho_L)$ when the equivalence bounds $(\rho_L, \rho_U)$ and other settings are fixed. When ρ is a constant, the sample size decreases with wider range of $(\rho_L, \rho_U)$. The computed sample sizes of the EQUT-$r$ procedure are slightly smaller than those of the EQUT-$\hat{\zeta}$ transformation for small ρ < 0.3. The situation is reversed when ρ = 0.4 with $(\rho_L, \rho_U)$ = (0.3, 0.5), and when ρ = 0.5 with $(\rho_L, \rho_U)$ = (0.4, 0.6). More importantly, the small discrepancy between the simulated power and estimated power reveals that the two techniques are extremely accurate for power and sample size calculations. In short, the extended sample correlation coefficient and Fisher's *z* transformation procedures can be recommended as general tools for appraising correlation equivalence.

## Conclusions

A growing attention in the behavioral and psychological literature concerns how to make a decision about an observed effect that is small enough to be considered negligible. However,

the conventional tests of difference are often inappropriately applied to conclude an effect is absent based a non-significant result. A widely recommended approach is to conduct an equivalence test to ascertain whether the observed effect size falls inside the selected equivalence boundaries. The TOST procedure of mean equivalence has been extensively applied in pharmacokinetics and various scientific disciplines. It is essential to note that there is little consensus in the literature on which method is most appropriate for equivalence testing. Conceptually, the preference varies with the right and proper criteria to select an optimal procedure. Considerations of more advanced aspects of TOST and alternative procedures for bioequivalence testing are beyond the scope of this article. The interested reader is referred to Meyners [24], Berger and Hsu [30], and the discussion therein for further details.

In view of the prevalent recognition of TOST, Goertzen and Cribbie [31] applied the same principle to the problem of assessing a lack of association. However, their numerical results showed that the TOST correlation procedure does not maintain nominal rejection rates when the sample sizes and correlation bounds are small. Despite the undesirable behavior of the TOST extension for correlation evaluations, no technical examinations and proper alternatives have been described in the literature. The present article aims to contribute to the correlation equivalence studies in four aspects. First, based on the Pearson product-moment correlation coefficient and the Fisher's $z$ transformation, their asymptotic properties are extended to construct equivalence procedures of correlation coefficients. Second, the empirical and analytic investigations not only clarify situations that the TOST principle does not adequately attain the nominal Type I error rates, but also justify the overall performance of the improved techniques for correlation assessments. Third, to enhance the utility of the suggested procedures, the corresponding power and sample size calculations for designing correlational research are also considered. Fourth, computer algorithms are developed to facilitate the practical use of the proposed equivalence procedures by providing efficient and accurate calculations of rejection regions, statistical powers, and sample sizes for correlation equivalence studies.

## Supporting information

**S1 File. R programs for performing the equivalence test procedures.**
(DOCX)

**S2 File. SAS/IML programs for performing the equivalence test procedures.**
(DOCX)

## Author Contributions

**Conceptualization:** Gwowen Shieh.

**Data curation:** Gwowen Shieh.

**Formal analysis:** Gwowen Shieh.

**Funding acquisition:** Gwowen Shieh.

**Investigation:** Gwowen Shieh.

**Methodology:** Gwowen Shieh.

**Project administration:** Gwowen Shieh.

**Resources:** Gwowen Shieh.

**Software:** Gwowen Shieh.

**Supervision:** Gwowen Shieh.

**Validation:** Gwowen Shieh.

**Visualization:** Gwowen Shieh.

**Writing – original draft:** Gwowen Shieh.

**Writing – review & editing:** Gwowen Shieh.

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
