## [Decision Letter · Decision Letter 0]

19 Mar 2021

PONE-D-21-04114

Improved procedures and computer programs for equivalence assessment of correlation coefficients

PLOS ONE

Dear Dr. Shieh,

Thank you for submitting your manuscript to PLOS ONE. After careful consideration, we feel that it has merit but does not fully meet PLOS ONE’s publication criteria as it currently stands. Therefore, we invite you to submit a revised version of the manuscript that addresses the points raised during the review process.

It  would be helpful if you could tone down the criticism of the work presented previously by Goertzen and Cribbie (2010) .  The main issue is the lack of a simulation power study.  Once that is completed and the other comments are attended to I believe this manuscript should be suitable for publication. 

We look forward to receiving your revised manuscript.

Kind regards,

Alan D Hutson

Academic Editor

PLOS ONE

Journal Requirements:

Reviewers' comments:

Reviewer's Responses to Questions

**Comments to the Author**

1. Is the manuscript technically sound, and do the data support the conclusions?

Reviewer #1: Partly

Reviewer #2: Yes

2. Has the statistical analysis been performed appropriately and rigorously? 

Reviewer #1: Yes

Reviewer #2: N/A

3. Have the authors made all data underlying the findings in their manuscript fully available?

Reviewer #1: Yes

Reviewer #2: Yes

4. Is the manuscript presented in an intelligible fashion and written in standard English?

Reviewer #1: Yes

Reviewer #2: Yes

5. Review Comments to the Author

Reviewer #1: You describe an extension to the paper by Goertzen and Cribbie on equivalence testing for a correlation coefficient. Goertzen and Cribbie (G&C) propose and evaluate Two-One-Sided-Tests (TOST) using either the asymptotic distribution of the correlation coefficient or the asymptotic distribution of the Fisher z transformed correlation coefficient. You critique the TOST because of its well known (demonstrated but not acknowledged explicitly by G&C) conservativeness when the standard error of the estimand is large or the equivalence interval is narrow. You apply less conservative methods developed in various places (e.g. Berger and Hsu 1996) and described in Wellek’s books (2nd ed. 2010). The paper is technically correct and the provided R and SAS code is practically useful.

My two major comments concern the use of the word “control” applied to the type I error rate of a conservative test and your omission of the reasons why practitioners continue to use the TOST.

You repeatedly describe the TOST as failing to control the type I error rate. In my opinion, this overstates the concern. It is clear that the TOST can be conservative; no one disputes that. I would not call that “failing to control” the type I error rate. Dictionary definitions of control include concepts of restraining, limiting and restricting. All these definitions concern imposing an upper bound. A liberal test, with an empirical type I error rate substantially exceeding the nominal rate, does fail to control. A conservative test does not fail to control. Your concerns about failing to control the type I error rate should be reworded as TOST is conservative and sometimes substantially so.

The differences between TOST and your less conservative method are found at small sample sizes (and when the variance is a free parameter, at large error variances). Your figure 4 demonstrates this (as does figure 3, but less clearly). Similar issues are found with traditional equivalence tests of differences of means of data with normally distributed errors. Those two figures demonstrate why some folks are reluctant to use a less conservative method. The critical region expands as the sample size drops below N=100. The probability content is the same (5%), but because the sampling distribution is more diffuse when N is smaller, the critical region has to expand. Using figure 3 as an example, an observed correlation of 0.012 would be judged ‘not equivalent to 0’ when N = 100 but judged ‘equivalent to 0’ with N = 30. Many practitioners find this unacceptable. Two papers that discuss this issue are Schuirmann’s discussion of the Berger and Hsu paper (1996, Stat Sci 11:283-319) and Meyners 2007,(Food Quality and Preference, DOI: 10.1016/j.foodqual.2006.07.005). I am not suggesting that a less conservative method is wrong. I am strongly suggesting that you acknowledge that there isn’t a clear favorite and that the less conservative methods are not universally favored.

You fault the TOST for not maintaining a specified type I error rate. You show situations where your proposed method does maintain the nominal type I error rate, but you only consider N down to 25. For sufficiently small N and narrow equivalence region, your methods also do not maintain the type I error rate. For example, consider N = 10 and equivalence bounds of (-0.01, 0.01). In this situation, the sampling distribution of r, in the most favorable situation of ρ=0, has only a 2.1% probability in the interval (-0.01, 0.01). I agree that this failure of your proposed method to maintain the nominal type I error rate only occurs at ridiculously small N and narrow equivalence regions, but you should not imply that the proposed method always maintains the specified type I error rate.

Details:

The focus on the main point can be improved by tightening up the writing throughout so that the paper. One large example is the discussion of the empty set critical region (Table 2 and associated discussion), which I found peripheral to the main points. A smaller example is the first 3 sentences of the first paragraph of the introduction. These talk about effect sizes, which is not the point of the paper.

Throughout, you should include specifics of pages or chapters when you cite books. You do this sometimes.

Reviewer #2: See attached......................................................................................................................................................................................................................................

6. PLOS authors have the option to publish the peer review history of their article (what does this mean?). If published, this will include your full peer review and any attached files.

Reviewer #1: No

Reviewer #2: No

---

## [Decision Letter · Decision Letter 1]

14 May 2021

Improved procedures and computer programs for equivalence assessment of correlation coefficients

PONE-D-21-04114R1

Dear Dr. Shieh,

We’re pleased to inform you that your manuscript has been judged scientifically suitable for publication and will be formally accepted for publication once it meets all outstanding technical requirements.

Kind regards,

Alan D Hutson

Academic Editor

PLOS ONE

Additional Editor Comments (optional):

Reviewers' comments:

Reviewer's Responses to Questions

**Comments to the Author**

1. If the authors have adequately addressed your comments raised in a previous round of review and you feel that this manuscript is now acceptable for publication, you may indicate that here to bypass the “Comments to the Author” section, enter your conflict of interest statement in the “Confidential to Editor” section, and submit your "Accept" recommendation.

Reviewer #1: All comments have been addressed

Reviewer #2: (No Response)

2. Is the manuscript technically sound, and do the data support the conclusions?

Reviewer #1: (No Response)

Reviewer #2: Yes

3. Has the statistical analysis been performed appropriately and rigorously? 

Reviewer #1: (No Response)

Reviewer #2: Yes

4. Have the authors made all data underlying the findings in their manuscript fully available?

Reviewer #1: (No Response)

Reviewer #2: Yes

5. Is the manuscript presented in an intelligible fashion and written in standard English?

Reviewer #1: (No Response)

Reviewer #2: Yes

6. Review Comments to the Author

Reviewer #1: Thank you for responding to my concerns about the wording of “controlling” the Type I error rate, the controversy about equivalence test methods, and the criticism of Goertzen and Cribben.

I also accept your claim that your procedure maintains the specified type I error rate. I note in passing (no revision needed) that this is because rejection regions for your test can be wider than the population equivalence region. Using the example in your reviewer response, the equivalence region is –0.01 < rho < 0.01, but the rejection region is -0.0232 < r < 0.0232. I suspect users of an equivalence test will be confused when their data says r = 0.02 but this is still considered as rejecting the null hypothesis that rho < -0.01 or rho > 0.01. This is one of the reasons for the difference of opinion about equivalence test methods: strict mathematical adherence to Neyman-Pearson principles or more common sense.

The manuscript is generally well written but should get one last read for clarity and use of English. For example, lines 3-4 are probably missing a word after recent.

Reviewer #2: The author was in general very clear and diligent with their revisions, and should be commended for this work. There is just one minor issue remaining before I believe the article is ready for publication:

** With regard to a justification for the proposed procedure, the authors send the reader to Wellek and Lehmann

& Romano. Neither of these is an easy read, and moreover, a clear justification/rationale for the proposed approach should be present in the manuscript. Thus, I am asking the author to summarize the material in Wellek and Lehmann & Romano in a paragraph that is easily accessible to applied researchers from the social/behavioral sciences;

Minor point: I believe the authors missed my point about Goertzen & Cribbie's finding that it is easy to get large correlation values just by chance with small N. The point was that this makes it hard to declare the relationship negligible (not different than 0) ... because can we really declare large correlations negligible?? The power results support this contention as power is really low at small N, even for the proposed procedure. Regardless ... these points do not impact the paper at all ... just a brief clarification.

7. PLOS authors have the option to publish the peer review history of their article (what does this mean?). If published, this will include your full peer review and any attached files.

Reviewer #1: No

Reviewer #2: No

---

## [Editor Report · Acceptance letter]

19 May 2021

PONE-D-21-04114R1 

Improved procedures and computer programs for equivalence assessment of correlation coefficients 

Dear Dr. Shieh:

I'm pleased to inform you that your manuscript has been deemed suitable for publication in PLOS ONE. Congratulations! Your manuscript is now with our production department. 

Kind regards, 

on behalf of

Dr. Alan D Hutson 

Academic Editor

PLOS ONE